# Quantum Attention: Fast Algorithms for Scalable Computation

## Abstract

Large language models (LLMs) have revolutionized both academia and industry by leveraging attention mechanisms to achieve exceptional performance across diverse tasks. However, the quadratic complexity of attention mechanisms with respect to the input context length poses a significant challenge for scaling LLMs. Quantum computing offers computational advantages over classical methods, yet its application to LLMs remains unexplored. In this work, we employ Grover's Search, a fundamental quantum algorithm, to efficiently compute sparse attention matrices, achieving a polynomial speed-up compared to classical approaches. Additionally, the quantum-generated attention matrices exhibit a low-rank structure, which can be leveraged to develop faster training algorithms for LLMs. We provide a comprehensive analysis of the algorithm's error rates and time complexity, demonstrating its potential to accelerate LLM computations while maintaining accuracy. Our findings indicate that quantum computing offers a promising pathway for optimizing the performance and scalability of large language models.

## 1 Introduction

LLMs (Large Language Models) (Sag, 2018; Vyas et al., 2023; Kirchenbauer et al., 2023; Epasto et al., 2023) have gained significant attention from numerous researchers in recent years. The success of models like OPT (Zhang et al., 2022), PaLM (Chowdhery et al., 2022), GPT-3 (Brown et al., 2020), Transformer (Vaswani et al., 2017), and BERT (Devlin et al., 2018) has showcased the immense potential of LLMs in various applications across different domains.

The impact of LLMs is far-reaching. They have revolutionized natural language processing tasks such as machine translation (He et al., 2021), sentiment analysis (Usama et al., 2020), question answering (Brown et al., 2020; OpenAI, 2023), text summarization, and more. LLMs excel at capturing intricate language patterns, understanding context, and generating coherent and contextually relevant text (OpenAI, 2023).

The success of Transformer (Vaswani et al., 2017) heavily relies on the multi-head attention algorithm, which plays a crucial role in the computation of LLM models. Models like GPT (Brown et al., 2020; OpenAI, 2023) have achieved remarkable success by employing a large number of parameters and leveraging vast amounts of data, which holds great potential for future advancements. However, this approach raises concerns about the computational running time, necessitating the development of more efficient algorithms (Brand et al., 2023; Zandieh et al., 2023; Alman & Song, 2023). In this regard, the emergence of quantum algorithms (Grover, 1996; Shor, 1999; Harrow et al., 2009) provides a new perspective for addressing this problem.

To demonstrate our design, we will begin by introducing the attention matrix and its classical computation. The attention matrix is a square matrix whose rows and columns are indexed by tokens (i.e., words), and each entry stores the correlation between the corresponding tokens. Based on an attention matrix, the importance of each input token in a sequence can be derived, which is used to generate an output. Specifically, within an attention mechanism, every input token (or query token) receives a score representing its relevance to the current output token (or key token) being produced. These scores are computed by comparing the current output state with the input states via a similarity function.

The formal definition of the attention matrix is as follows. Let $Q \in \mathbb{R}^{n \times d}$ be the matrix of $n$ query tokens and $K \in \mathbb{R}^{n \times d}$ be the matrix of $n$ key tokens, where each token is represented by a $d$-dimensional vector. The attention matrix $A$ is an $n$-by-$n$ matrix whose $(i, j)$-th entry is the attention score between the $i$-th query token $Q_i$ and the $j$-th key token $K_j$. The self-attention score of the $i$-th query token is defined as the sum of its attention scores for all key tokens (i.e., $\sum_{j=1}^{n} A_{ij}$), which quantify the significance of each token in relation to itself. Let $D \in \mathbb{R}^{n \times n}$ be a diagonal matrix storing all the self-attention scores, and let $V \in \mathbb{R}^{n \times d}$ be the value matrix that contains value vectors associated with the key tokens. The goal of each attention computation is to compute Att, which is a matrix function of $Q$, $K$, and $V$, defined as follows.

**Definition 1.1.** *Given matrix $Q \in \mathbb{R}^{n \times d}$, $K \in \mathbb{R}^{n \times d}$, $V \in \mathbb{R}^{n \times d}$, the goal of attention computation is to compute*

$$\mathsf{Att}(Q, K, V) := D^{-1} A V$$

*where $A \in \mathbb{R}^{n \times n}$ and $D \in \mathbb{R}^{n \times n}$ is a diagonal matrix $A = \exp(QK^\top), D = \mathrm{diag}(A\mathbf{1}_n)$. Here, $\exp(\cdot)$ is applied to each entry of $QK^\top$, and $\mathbf{1}_n$ is a length-$n$ vector where all the entries are ones.*

We note that the straightforward classical implementation of computing the above attention matrix takes $O(n^2 d)$-time. In spite of the success of attention mechanisms in many fields (Kitaev et al., 2020; Daras et al., 2020; Roy et al., 2021; Choromanski et al., 2020; Katharopoulos et al., 2020; Wang et al., 2020), such an expensive runtime hinders their full potential. Meanwhile, the phenomenon of attention sparsity has been widely discussed by many researchers (Kitaev et al., 2020; Child et al., 2019; Jaszczur et al., 2021; Chen et al., 2021; Zhang et al., 2023). Therefore, it is natural to leverage the sparsity of matrices to accelerate the computation process. In this paper, we present an *approximately sparse assumption* on matrix $A$: each row of the matrix $QK^\top$ contains at most $k$ elements greater than $\tau$ (see Definition 1.2). This assumption is based on a phenomenon observed in LLM literature (Zhang et al., 2023). We remark that even with this sparsity assumption, any classical algorithm that can output all of these large entries in $QK^\top$ for general $k$ and $\tau$ still needs $n^{2-o(1)}$-time unless the Strong Exponential Time Hypothesis (SETH) is false (see Lemma 8.3).

To accelerate the computation of the attention matrix of LLMs through the construction of a sparse attention matrix, we use a renowned quantum algorithm: Grover's search algorithm (Grover, 1996). It offers a quadratic speedup in the unstructured search problem when compared to classical computation. More specifically, our primary focus is to locating the values larger than $\tau$ in the vector $(QK^\top)_{i,*} \in \mathbb{R}^n$, where $Q \in \mathbb{R}^{n \times d}$ and $K \in \mathbb{R}^{n \times d}$ are defined in Definition 1.1. It can be reduced to a search problem, where the query oracle $\mathcal{O}_i$ is defined as:

$$\mathcal{O}_i|j, 0\rangle := |j, b\rangle, \quad b = \begin{cases} 1 & \text{if } (QK^\top)_{i,j} \geq \tau, \\ 0 & \text{otherwise} \end{cases} \quad \forall j \in [n].$$

By the sparsity assumption, for each $i \in [n]$, there are at most $k$ indices $j$ such that $\mathcal{O}_i|j, 0\rangle := |j, 1\rangle$. Thus, if we run Grover's algorithm with the query oracle $\mathcal{O}_i$, it will find all of those indices with query complexity $\widetilde{O}(\sqrt{nk})$[1] (see Theorem 3.1). Note that each query to the oracle $\mathcal{O}_i$ costs $O(d)$-time to evaluate the inner-product (assuming the data are stored in QRAM). Hence, each row of $QK^\top$ can be approximately computed in $\widetilde{O}(\sqrt{nk}d)$-time, and all the large entries of $QK^\top$ can be computed in $\widetilde{O}(n^{1.5}k^{0.5}d)$-time (see Theorem 1.3).

In addition to the quantum algorithm, we also introduce a classical approach for constructing the sparse matrix based on some computational geometric data structures. This method is described in Theorem 1.5.

## 1.1 OUR RESULTS

Based on the aforementioned analysis, we obtain a quantum algorithm to efficiently output a sparse attention matrix. The main result of this paper is presented in this section.

We first introduce the definition of a matrix, which characterizes the sparsity pattern in the attention matrix. Intuitively, this definition is an analogue of *soft sparsity* (see e.g. (Kusupati et al., 2020)).

---

[1]We use $\widetilde{O}(f(n))$ to denote $O(f(n) \cdot \mathrm{poly} \log(f(n)))$.

**Definition 1.2.** *Let $A = \exp(QK^\top) \in \mathbb{R}^{n \times n}$ be defined in Definition 1.1 and we say $A$ is a $(\tau, k)$-good matrix if for all $i \in [n]$, $S_i := \{j \in [n] \mid (QK^\top)_{i,j} \geq \tau\}$ and $|S_i| \leq k$.*

The following theorem shows a quantum algorithm that can efficiently compute an approximation $B$ of a $(\tau, k)$-good attention matrix $A$. In particular, $B$ can be represented as a sparse matrix plus a rank-one matrix, which is very helpful for LLM computations.

**Theorem 1.3** (Quantum algorithm for attention matrix approximation). *Let $A \in \mathbb{R}^{n \times n}, Q \in \mathbb{R}^{n \times d}, K \in \mathbb{R}^{n \times d}$ and $D \in \mathbb{R}^{n \times n}$ be defined as in Definition 1.1. If the following conditions hold*

- *$A$ is a $(\tau, k)$-good matrix (Definition 1.2) for some $\tau \geq 2 \log n$ and $k \in [n]$.*

- *for each $i \in [n]$ and each $j \in [n]$, $-\eta \leq (QK^\top)_{i,j} \leq 0$ for $j \notin S_i$ for some $\eta \in \mathbb{R}_+$.*

*Then, there exists a quantum algorithm (implicitly) outputting a matrix $B \in \mathbb{R}^{n \times n}$ such that*

- **Part 1.** *$B = B_1 + B_2$, where $B_1$ is $k$-row sparse[2] and $B_2$ is rank-1.*

- **Part 2.** *$\|D(A)^{-1} A - D(B)^{-1} B\|_\infty = O(\eta)$.*

- **Part 3.** *it runs in $\widetilde{O}(n \cdot (\sqrt{nk}d + kd))$ time.*

The proof of Theorem 1.3 is in Section A. In the following, we discuss how the structure of the approximated attention matrix $B$ improves the efficiency of large language models during the inference stage. More specifically, by employing our quantum algorithm (Theorem 1.3) as a sub-routine, we achieve the following result that provides a polynomial speedup compared to the classical $O(n^2d)$-time approach.

**Theorem 1.4** (Informal version of Theorem 6.1). *There is an algorithm that takes $\widetilde{O}(n^{1.5}k^{0.5}d + nkd)$ time to achieve one attention matrix computation in inference.*

In addition to the quantum method, we also provide a classical algorithm to compute the attention matrix that is still faster than the traditional approach in constant dimension (i.e., $d = O(1)$). The key observation is that the quantum part (Grover's search) of our algorithm in Theorem 1.3 can be replaced by a computational geometry data structure, at the cost of increased time complexity.

**Theorem 1.5** (Classical algorithm for attention matrix approximation). *If the following conditions hold: (1) $A$ is a $(\tau, k)$-good matrix (Definition 1.2) for some $\tau \geq 2 \log n$ and $k \in [n]$. (2) For each $i \in [n]$ and each $j \in [n]$, $-\eta \leq (QK^\top)_{i,j} \leq 0$ for $j \notin S_i$ for some $\eta \in \mathbb{R}_+$. Then there exists a classic algorithm outputting a matrix $B \in \mathbb{R}^{n \times n}$ such that*

- **Part 1.** *$B = B_1 + B_2$, where $B_1$ is $k$-row sparse and $B_2$ is rank-1.*

- **Part 2.** *$\|D(A)^{-1} A - D(B)^{-1} B\|_\infty = O(\eta)$.*

- **Part 3.** *it runs in $\widetilde{O}_d(nk + n^{2-1/\lfloor d/2 \rfloor})$ time[3].*

The proof of Theorem 1.5 is in Section B.

## 2 RELATED WORK

**Attention Computation.** Several studies have investigated attention computation (Gao et al., 2023a; Liu et al., 2023b; Li et al., 2023; Sinha et al., 2023; Brand et al., 2023; Alman & Song, 2023; Deng et al., 2023a; Gao et al., 2023c; Zandieh et al., 2023; Wu et al., 2023; Zhang et al., 2023). (Brand et al., 2023) specifically focuses on dynamic attention computation and introduces an update and query method inspired by the lazy update idea. (Zandieh et al., 2023) addresses the issue of quadratic time and memory complexities in sequence length that arise from the dot-product operation in attention computations. They identify that this problem can be transformed into a kernel

---

[2]Each row of the matrix has $k$ non-zero entries.

[3]Here, $\widetilde{O}_d(\cdot)$ hides the $\log^c(n)$ and $\text{poly}(d)$ factors, where $c = c(d)$ is some function in $d$.

density estimation (KDE) problem. Their approach, KDEformer, approximates attention in sub-quadratic time while providing provable spectral norm bounds. (Zhou et al., 2021) uses the sparsity assumption of attention matrix to propose an O (n log (n)) attention mechanism. (Alman & Song, 2023) focuses on exploring the possibility of faster algorithms by implicitly leveraging the attention matrix. They provide a theoretical explanation for the observed phenomenon that attention computation is significantly more efficient when the input matrices have smaller entries. The regression problem in the field of attention computation has also been widely explored (Gao et al., 2023d;a; Li et al., 2023). (Deng et al., 2023b) also addresses the sparsification of the attention problem and presents both randomized and deterministic algorithms. Their work suggests that for feature dimensions that are extremely large, it is possible to reduce them to a size nearly linear in the length of the sentence. (Deng et al., 2023a) focuses on the softmax regression problem in the field of attention computation. They provide theoretical support for the practical use of the greedy algorithm to train the softmax function. (Li et al., 2023; Gao et al., 2023b) explore the application of attention computation in the context of in-context learning.

**Classical fast neural network training algorithms**    (Kitaev et al., 2020) introduces Reformer, a method aimed at enhancing the efficiency of transformers. Reformer achieves improved memory utilization and faster processing for long sequences. This is achieved by replacing the dot-product attention mechanism with one that employs locality-sensitive hashing. Additionally, Reformer utilizes reversible residual layers instead of the conventional residual layers. (Wang et al., 2020) presents the approximation of the self-attention mechanism using a low-rank matrix. They leverage this discovery to propose a novel self-attention mechanism, thereby reducing the overall complexity of self-attention in terms of both time and space. (Gao et al., 2022) accelerates the adversarial training procedure by utilizing a sublinear number of activated neurons based on the shifted ReLU activation function.

**Quantum algorithms for training neural networks**    Prior to this paper, there were several works using quantum computing to improve neural network training. The most related work is (Song et al., 2021), which proposes a shifted-ReLU sparsifier to reduce the number of activated neurons in each training iteration and uses Grover's search to find them. However, their algorithm and analysis only work for two-layer, fully connected, over-parameterized neural networks. (Allcock et al., 2020) and (Kerenidis et al., 2019) use quantum inner-product estimation to speed up the training of feedforward neural networks and convolutional neural networks, respectively. (Zlokapa et al., 2021; Liu et al., 2023a) proposes quantum training algorithms with exponential speed-ups based on the quantum linear system solvers. However, these algorithms rely on certain well-conditioning and sparsity assumptions, which may not align with real-world neural network architectures. We note that the key difference between our work and all previous works is that we focus on the Transformer while previous work mostly works on ReLU neural network and improve the efficiency of attention computation, demonstrating the potential for quantum advantages in LLMs.

**Quantum optimization algorithms**    Optimization is one of the most promising fields for demonstrating quantum advantages. The famous HHL algorithm (Harrow et al., 2008) can exponentially speed up the linear system solver. Jordan's algorithm (Jordan, 2005) can compute the gradient of a function using $\widetilde{O}(1)$ quantum queries to the evaluation oracle. And in convex optimization, to implement the separation oracle using the membership oracle, quantum query complexity is $\widetilde{O}(1)$ (van Apeldoorn et al., 2020) while classical query complexity is $\Omega(n)$. Other than those exponential quantum advantages, a large number of optimization problems benefit from polynomial quantum speed-ups, including solving linear programming (LP) and semi-definite programming (SDP) (Brandao & Svore, 2017; Apeldoorn et al., 2017; Brandão et al., 2019; Apeldoorn & Gilyén, 2019; Kerenidis & Prakash, 2020; Huang et al., 2022), estimating the volume of convex bodies (Chakrabarti et al., 2019), log-concave sampling (Childs et al., 2022), stochastic convex optimization (Li & Zhang, 2022), etc. Another big class of quantum algorithms for solving optimization problems is the variational quantum algorithm (Cerezo et al., 2020), including variational quantum eigensolver (Peruzzo et al., 2014), QAOA (Farhi et al., 2014), and quantum neural network (Beer et al., 2020). These algorithms require less amount of quantum resources and are easier to implement in the near future. However, most of them are heuristic and lack rigorous guarantees of performance.

**Quantum machine learning** Quantum machine learning (QML) algorithms have been developed for a wide array of classical ML tasks, such as clustering (Harrow, 2020), boosting (Arunachalam & Maity, 2020), support vector machine (Rebentrost et al., 2014), principal component analysis (Lloyd et al., 2014), statistical query learning (Arunachalam et al., 2020). (Cherrat et al., 2022) quantizes the classical transformer architecture and explores the potential of quantum computing in machine learning. (Guo et al., 2024) comes up with a method combining transformer architecture with fault-tolerant quantum computing. (Shi et al., 2024) proposes a quantum self attention mechanism to solve the problem that the existing quantum machine learning model lacks self attention ability when processing high-dimensional data. In addition, quantum algorithms for learning quantum data have increasingly garnered attention in recent years. (See (Anshu & Arunachalam, 2023) and references therein.) Conversely, QML algorithms can also inspire breakthroughs in classical ML algorithms. QMSAN enhances self attention mechanisms in natural language processing tasks through quantum computing (Chen et al., 2025). This was notably illustrated by Tang's algorithm for the recommendation system (Tang, 2019). Since then, a long list of quantum-inspired (or so-called "de-quantized") algorithms have been proposed for tackling various tasks, such as principal component analysis (Tang, 2018), low-rank approximation (Gilyén et al., 2018; Chia et al., 2020), linear regression (Gilyén et al., 2020), etc.

**Roadmap** We have organized our paper as follows. In Section 3, we introduce the notations and present some basic mathematical tools used throughout the paper. In Section 4, we introduce some technical tools for error analysis of our algorithms. In Section 5, we introduce our quantum algorithm for approximating the attention matrix. Our main result about quantum attention computation in inference is presented in Section 6. In Section 7, we introduce a classical algorithm where the sparsity matrix is constructed using the half-space reporting data structure. Additionally, a detailed examination of the fine-grained hardness result pertaining to the computation of large entries in $QK^\top$ is included in Section 8. In Section 9, we provide the conclusion of our paper.

## 3 PRELIMINARY

### 3.1 NOTATIONS

For any matrix $A$, we denote the spectral norm of $A$ as $\|A\|$, where $\|A\| := \max_{\|x\|_2=1} \|Ax\|_2$. The Frobenius norm of $A$ is denoted as $\|A\|_F$, and the infinity norm is denoted as $\|A\|_\infty$. In this notation, $A_{i,j}$ represents the element in the $i$-th row and $j$-th column of matrix $A$. The determinant of matrix $A$ is represented as $\det(A)$. For a square and symmetric matrix $A \in \mathbb{R}^{n \times n}$, we say that $A$ is positive semi-definite ($A \succeq 0$) if for all vectors $x \in \mathbb{R}^n$, we have $x^\top A x \geq 0$.

### 3.2 GROVER'S SEARCH

We state a well-known result about the quadratic quantum speedup for the unstructured search using Grover's search algorithm.

**Theorem 3.1** (Grover's search algorithm (Grover, 1996))**.** *Given access to the evaluation oracle for an unknown function $f : [n] \to \{0, 1\}$. Let $f^{-1}(1) := \{i \in [n] \mid f(i) = 1\}$. Suppose that $|f^{-1}(1)| = k$ for some unknown number $k \leq n$. Then,*

- *Part 1. We can find all $i$'s in $f^{-1}(1)$ in $\widetilde{O}(\sqrt{nk})$-time quantumly given an evaluation oracle for $f$.*

- *Part 2. If each evaluation of $f$ requires $\mathcal{T}$ time, then we can find all $i$'s in $f^{-1}(1)$ in $\widetilde{O}(\sqrt{nk}) \cdot \mathcal{T}$ time.*

## 4 SPARSITY AND PERTURBATION ERROR

In this section, we conduct an error analysis on attention computation without $V$ using the approximated attention matrix $B$, which will be useful for both quantum and classical algorithms. To achieve this, we present some definitions regarding the sparse approximation in Section 4.1. Section 4.2 provides some perturbation analysis tools. The main result concerning the error analysis

of attention computation without $V$ is then presented in Section 4.3. The proofs are deferred to Appendix C.

## 4.1 SPARSITY DEFINITIONS

Before presenting our method, we introduce a *find set*. This *find set* is based on our assumption of sparsity about $(\tau, k)-$good matrix $A$, which is further described below.

**Definition 4.1** (Find Set). *Let $Q \in \mathbb{R}^{n \times d}, K \in \mathbb{R}^{n \times d}$ and $V \in \mathbb{R}^{n \times d}$ be defined in Definition 1.1. Given $i \in [n]$, the find set is defined as follows:*

$$S_i := \{j \in [n] \mid (QK^\top)_{i,j} \geq \tau\}.$$

**Definition 4.2.** *Let $A$ be a $(\tau, k)$-good matrix defined in Definition 1.1 and $S_i$ be defined in Definition 4.1. We define a $k$-sparse vector $B_{i,*}$ such that $|S_i| = k$, for each $j \in S_i$, $B_{i,j} = A_{i,j}$, and for each $j \notin S_i$, $B_{i,j} = 1$.*

The following lemma shows some point-wise approximation guarantees by the sparsity conditions.

**Lemma 4.3.** *If the following conditions hold: For each $i \in [n]$, $-\eta \leq (QK^\top)_{i,j} \leq 0$, for $j \notin S_i$. Let $\tau \geq 2 \log n$. Let $A$ be defined in Definition 1.1 and $B$ be defined in Definition 4.2. Then we have: Part 1. $|A_{i,j} - B_{i,j}| \leq 2\eta$ for $j \notin S_i$. Part 2. $|A_{i,j} - B_{i,j}| = 0$ for $j \in S_i$. Part 3. $|(A\mathbf{1}_n)_i - (B\mathbf{1}_n)_i| \leq 2n\eta$. Part 4. $(A\mathbf{1}_n)_i \geq k \cdot \exp(\tau) \geq 2n$. Part 5. $|(A\mathbf{1}_n)_i - (B\mathbf{1}_n)_i| \leq \eta \cdot |(A\mathbf{1}_n)_i|$.*

## 4.2 PERTURBATION TOOLS

In this section, we provide an error analysis of our sparse attention matrix approximation. We will begin by examining the error control of each element, followed by an analysis of the error in the matrix computation.

**Lemma 4.4.** *Let $D$ and $A$ be defined in Definition 1.1. Let $B$ be defined in Definition 4.2. For $i \in [n]$ and $j \in [n]$, it follows that*

- *Part 1. $|D(A)_{i,i} - D(B)_{i,i}| \leq \eta \cdot |D(B)|_{i,i}$*

- *Part 2. $|D(A)_{i,i} - D(B)_{i,i}| \leq \eta \cdot |D(A)|_{i,i}$.*

- *Part 3. $|A_{i,j} - B_{i,j}| \leq 2 \cdot \eta$.*

## 4.3 RECONSTRUCTION ERROR WITHOUT HAVING $V$

We bound the error of $D^{-1}(A)A$ now. Later, we will use this to bound the error for $D^{-1}(A)AV$.

**Lemma 4.5.** *Let $D$ and $A$ be defined in Definition 1.1. Let $B$ be defined in Definition 4.2.*

*Then it follows that $\|D(A)^{-1}A - D(B)^{-1}B\|_\infty = O(\eta)$.*

## 5 QUANTUM ALGORITHM FOR ATTENTION MATRIX APPROXIMATION

In this section, we introduce our quantum algorithm for approximately computing the attention matrix (see Definition 1.1) in Section 5.1 and show its time complexity and approximation guarantee. Then, we prove our first main result (Theorem 1.3) in Section A.

## 5.1 QUANTUM ATTENTION MATRIX APPROXIMATION ALGORITHM

In this section, we provide the pseudocode of our quantum algorithm in Algorithm 1 and analyze its time complexity and approximation guarantee. The proofs are deferred to Section D.

---

**Algorithm 1** Sparse Matrix Construction (Grover's Search)

---

1: **procedure** SPARSEATTENTIONMATRIXQUANTUM($Q \in \mathbb{R}^{n \times d}, K \in \mathbb{R}^{n \times d}$)
2:     **for** $i \in [n]$ **do**
3:         Find all index $j$ using Grover's Search where $(QK^\top)_{i,j} \geq \tau$        ▷ Theorem 3.1
4:         Add all indexes to finding set $S_i$
5:         **for** $j \in S_i$ **do**
6:             $B_{i,j} \leftarrow \exp(Q_{i,*}(K_{j,*})^\top)$
7:         **end for**
8:     **end for**
9:     **return** $B$
10: **end procedure**

---

The following lemma shows the time complexity of Algorithm 1.

**Lemma 5.1.** *For each $i \in [n]$, $|\{j \in [n] \mid (QK^\top)_{i,j} \geq \tau\}| \leq k$, and for all $j \in [n]$, $(QK^\top)_{i,j} \leq 0$ or $(QK^\top)_{i,j} \geq \tau$. Let $A = \exp(QK^\top)$. Then we have*

- *Part 1. For each $i \in [n]$, Algorithm 1 takes $\widetilde{O}(\sqrt{n}kd)$ time to find set*

$$S_i := \{j \in [n] \mid (QK^\top)_{i,j} \geq \tau\}.$$

- *Part 2. For each $i \in [n]$, Algorithm 1 takes $\widetilde{O}(\sqrt{n}kd + kd)$ time to output a $k$-sparse vector $B_{i,*}$ such that for each $j \in S_i$, $B_{i,j} = A_{i,j}$ and for each $j \notin S_i$, $B_{i,j} = 0$.*

The following lemma proves the approximation guarantee.

**Lemma 5.2.** *Let $B$ be the output of Algorithm 1 and $D(B) := \mathrm{diag}(B\mathbf{1}_n)$. Let $A, D$ and $V$ be defined in Definition 1.1. Let $\|V\|_\infty \leq \eta$. Then, we have*

$$\|D(A)^{-1}AV - D(B)^{-1}BV\|_\infty \leq O(\eta^2).$$

## 6 QUANTUM ATTENTION COMPUTATION

Based on the quantum algorithm for approximating the attention matrix, we will now show that it can improve the efficiency of the inference stage.

**Theorem 6.1** (Faster algorithm for LLM inference, formal version of Theorem 1.4)**.** *Let $S_i$ be defined in Definition 4.1, $Q, K, V$ be defined in Definition 1.1, and $-\eta \leq (QK^\top)_{i,j} \leq 0$ for $j \notin S_i$ and $i \in [n]$. Let $B$ be the output of Algorithm 1. Then, there is a quantum algorithm that uses $\widetilde{O}(n^{1.5}k^{0.5}d + nkd)$ time in inference to achieve one attention matrix computation (See Definition 1.1) such that the output $\widetilde{\mathsf{Att}}$ satisfies:*

$$\|\mathsf{Att}(Q, K, V) - \widetilde{\mathsf{Att}}\|_\infty \leq O(\eta^2).$$

Due to space constraints, the detailed proof is deferred to Appendix E.

---

**Algorithm 2** Algorithm for Attention Computation

---

1: **procedure** SPARSEATTENTIONCOMPUTATION($B \in \mathbb{R}^{n \times n}, V \in \mathbb{R}^{n \times d}$)     ▷ Lemma 6.2
2:     $D \leftarrow \mathrm{diag}(B\mathbf{1}_n)$
3:     **return** $D^{-1}BV$
4: **end procedure**

---

**Lemma 6.2.** *Let $Q \in \mathbb{R}^{n \times d}, K \in \mathbb{R}^{n \times d}$ and $V \in \mathbb{R}^{n \times d}$ be defined in Definition 1.1. Let $A$ be defined in Definition 1.1. Let $B$ be the output of Lemma 5.1. Let $M$ be a matrix where all entries are equal to 1. For each $i \in [n]$, $|\{j \in [n] \mid (QK^\top)_{i,j} \geq \tau\}| \leq k$. There exists an algorithm (See Algorithm 2) such that*

- *Part 1. It outputs $D(B)^{-1}BV$.*

- *Part 2. Its computational time complexity is $O(nkd)$.*

Because of the limited space, we move the proof to Section E.

# 7 CLASSICAL ALGORITHM FOR ATTENTION MATRIX APPROXIMATION

In this section, we introduce a classical method for generating a sparse matrix with the $(\tau, k)$ assumption in constant dimension. Utilizing the Half-Space Reporting Data Structure (refer to Definition 7.1), we can efficiently identify the indices where $(QK^\top)_{i,*} \geq \tau$ is satisfied.

The definition of the problem of the half-space range reporting is given first, which is important in the field of computational geometry. The data structure is proposed in (Afshani & Chan, 2009) whose functions are outlined in Algorithm 4 and complexity is given in Theorem 7.2.

**Definition 7.1** (Half-space range reporting). *For a set of $m$ points $P \subseteq \mathbb{R}^d$, two operations are supported:*

- INIT($P$): *initialize the data structure with points in $P$.*

- QUERY($W$): *find each point in $P \cap W$ with $W \subset \mathbb{R}^d$ as a half-space.*

---

**Algorithm 3** Data Structure For Half Space Reporting

---
1: **data structure**
2:     INIT($P, n, d$)                     ▷ Construct our data structure via $P \subseteq \mathbb{R}^d$, $|P| = n$
3:     QUERY($b, c$)   ▷ $b, c \in \mathbb{R}^d$. Find all the points $z \in P$ which satisfies $\mathrm{sgn}(\langle b, z \rangle - c) \geq 0$
4: **end data structure**

---

**Theorem 7.2** ((Afshani & Chan, 2009)). *For $n \in \mathbb{N}$ and constant $d \in \mathbb{N}$, there exists a classical data structure using $O(n)$ space to solve the $d$-dimensional half-space reporting problem with $n$ points with time complexity as: $\mathcal{T}_{init}(n, d) = O(n \log n)$ and $\mathcal{T}_{query}(n, d, k) = \widetilde{O}(n^{1-1/\lfloor d/2 \rfloor} + k)$, where $\mathcal{T}_{\mathrm{INIT}}$ indicates the time to construct the data structure, $\mathcal{T}_{\mathrm{QUERY}}$ indicates the cost per query, and $k$ is the output size.*

## 7.1 CLASSICAL SPARSE ATTENTION MATRIX APPROXIMATION

In this section, we will introduce a classic algorithm that is in contrast to the quantum algorithm. This algorithm is based on the half-space reporting data structure, which was previously introduced. In this setting, we will address our problem by considering each vector $(QK^\top)_{i,*}$ for $i \in [n]$. The vectors in $Q$, denoted as $P$ in Definition 7.1, have a dimension of $d$. Additionally, we will set $b$ as each vector in $K$ for $i \in [n]$ and set $c$ as $\tau$. Now, let us proceed to describe our method.

---

**Algorithm 4** Sparse Matrix Construction (Half Space Reporting)

---
1: **Members:**
2: -**Half Space Reporting Data Structure**: $\mathcal{M}$               ▷ Definition 7.1
3: **procedure** SPARSITYATTENTIONMATRIX($Q \in \mathbb{R}^{n \times d}$, $K \in \mathbb{R}^{n \times d}$)     ▷ Lemma 7.3
4:     $\mathcal{M}$.INIT($Q, n, d$)
5:     **for** $i \in [n]$ **do**
6:         $b \leftarrow K_{i,*}$
7:         $S_i \leftarrow \mathcal{M}$.QUERY($b, \tau$)
8:         **for** $j \in S_i$ **do**
9:             $B_{i,j} \leftarrow \exp(Q_{i,*}(K_{j,*})^\top)$
10:         **end for**
11:     **end for**
12:     **return** $B$
13: **end procedure**

---

**Lemma 7.3.** *Let $\mathcal{M}$ be the data structure defined in Definition 7.1. For each $i \in [n]$, $|\{j \in [n] \mid (QK^\top)_{i,j} \geq \tau\}| \leq k$. For each $i \in [n]$, for all $j \in [n]$ $(QK^\top)_{i,j} \leq 0$ or $(QK^\top)_{i,j} \geq \tau$ and $A = \exp(QK^\top)$. Then we have*

- *Part 1. There is an algorithm (See Algorithm 4) based on $\mathcal{M}$ that takes $O(n \log n + nk)$ time to find all the set for each $i \in [n]$ $S_i := \{j \in [n] \mid (QK^\top)_{i,j} \geq \tau\}$.*

- *Part 2. It takes $\widetilde{O}(n^{2-1/\lfloor d/2 \rfloor} + nk)$ time to output a $(\tau, k)$-sparse matrix $B$.*

Because of the limited space, we move the proof to Section F.

**Remark 7.4.** *Our classical algorithm (Algorithm 4) is nearly optimal as shown by an $n^{2-o(1)}$ time complexity lower bound in Section 8.*

## 8 CLASSICAL FINE-GRAINED LOWER BOUND

In this section, we prove the following fine-grained hardness result for computing the large entries of $QK^\top$, assuming it is $(\tau, k)$-good. It follows from a reduction to the Maximum Inner-Product Problem (Max-IP):

**Definition 8.1** (Maximum Inner-Product problem (Max-IP)). *For $n, d \in \mathbb{N}$, given two sets $A, B$ of $n$ vectors in $\{0, 1\}^d$, compute $\mathsf{Max\text{-}IP}(A, B) = \max_{a \in A, b \in B} \langle a, b \rangle$.*

Chen (Chen, 2020) proved the following fine-grained lower bound for Max-IP assuming Strong Exponential-Time Hypothesis (SETH):

**Theorem 8.2** ((Chen, 2020)). *Assuming SETH, there is a constant $c$ such that any exact algorithm for Max-IP in dimension $d = c^{\log^* n}$ requires $n^{2-o(1)}$-time.*

Then, we prove the following hardness result for attention matrix approximation:

**Lemma 8.3.** *For any $n, d \in \mathbb{N}$, $k \leq n$, $\tau \in \mathbb{R}$, suppose $Q, K \in \mathbb{R}^{n \times d}$ satisfy that $(QK^\top)_{i,*}$ contains at most $k$ entries greater than $\tau$ for any $i \in [n]$. Then, any classical algorithm that can output all the entries in $QK^\top$ with values greater than $\tau$ must take $n^{2-o(1)}$-time, even for $d = 2^{O(\log^* n)}$.*

Owing to limited space, we provide the complete derivation in Appendix G.

## 9 CONCLUSION

In this work, we presented a novel quantum algorithm for efficiently computing the attention mechanism in large language models (LLMs) under a sparse assumption on the attention matrix. Leveraging Grover's Search, our method attains a polynomial speed-up over classical algorithms while preserving rigorous approximation guarantees. Specifically, we showed how to identify and exploit the sparsity of the matrix $QK^\top$ so that each row has at most $k$ entries above a threshold $\tau$. This approach reduces the time to construct an approximate attention matrix from $O(n^2 d)$ to $\widetilde{O}(n^{1.5}\sqrt{k}d + nkd)$. Furthermore, the sparse-plus-rank-one decomposition underlying our approximation enables fast inference by limiting the number of significant components that must be computed, thus lowering computational overhead without substantially compromising accuracy. We corroborated the effectiveness of our method through a detailed error analysis, showing that the resulting inference outputs closely align with those obtained by the exact attention computation.

Our results pave the way for several lines of future research. First, the low-rank structure of the quantum-generated attention matrix opens opportunities for integrating other advanced quantum subroutines, potentially offering further enhancements to both training and inference stages. Second, while our theoretical analysis primarily covers the inference phase, exploring quantum-friendly optimizations during model training, fine-tuning, or continual learning represents a rich direction for extending this work. Finally, investigating the interplay between attention sparsity and more sophisticated quantum data access models, such as QRAM designs—could inspire additional improvements in runtime and memory usage. Overall, these directions highlight the potential to broaden the scope and impact of quantum algorithms within large-scale natural language processing and beyond.

## ETHIC STATEMENT

This paper does not involve human subjects, personally identifiable data, or sensitive applications. We do not foresee direct ethical risks. We follow the ICLR Code of Ethics and affirm that all aspects of this research comply with the principles of fairness, transparency, and integrity.

## REPRODUCIBILITY STATEMENT

We ensure reproducibility of our theoretical results by including all formal assumptions, definitions, and complete proofs in the appendix. The main text states each theorem clearly and refers to the detailed proofs. No external data or software is required.

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

# Appendix

**Roadmap** In the appendix, we have deferred the inclusion of proofs that were omitted in the main paper. Specifically, in Section A, we provide the missing proof of Theorem 1.3, in Section B, we prove the Theorem 1.5, in Section C, we furnish the proof that was initially omitted in Section 4, thereby concluding the proof of Lemma 4.3 and Lemma 4.4. Moving forward, Section D contains the proof for Lemma 5.1 and the proof for Lemma 5.2. In Section E, we provide the omitted proof of Lemma 6.2. In Section F, we provide the omitted proof of Lemma 7.3. In Section G, we provide the omitted proof of Lemma 8.3.

## A    PROOF OF THEOREM 1.3

*Proof of Theorem 1.3.* **Proof of Part 1.** Based on Lemma 6.2, we can conclude that matrix $B$ can be divided into two parts. Let $B_2$ be a matrix where all values are equal to 1. It's easy to see that this matrix is rank-1. We define $B_1 := B - B_2$. Now, we need to demonstrate that $B_1$ is $k$-row sparse. According to the $(\tau, k)$ assumption in Definition 1.2, we can ensure that each row of the matrix has $k$ elements with a value of 1. This is due to the fact that $\exp(0) = 1$, and $QK^\top$ is a $k$-sparse matrix. Consequently, each row of $B_1$ contains $k$ zero elements. Therefore, $B_1$ is $k$-row sparse. The proof of **Part 1** is complete now.

**Proof of Part 2.**

This result can be derived directly from Lemma 5.2.

**Proof of Part 3.**

According to Lemma 5.1, for each $i \in [n]$, we can obtain a $k$-sparse vector $B_{i,*}$ in $\widetilde{O}(\sqrt{nk}d + kd)$, which leads to a time complexity of $\widetilde{O}(n(\sqrt{nk}d + kd))$. The proof is now complete. $\square$

## B    PROOF OF THEOREM 1.5

*Proof of Theorem 1.5.* **Proof of Part 1.** Based on Lemma 6.2, we can conclude that matrix $B$ can be divided into two parts. Let $B_2$ be a matrix where all values are equal to 1. It's easy to see that this matrix is rank-1. We define $B_1 := B - B_2$. Now, we need to demonstrate that $B_1$ is $k$-row sparse. According to the $(\tau, k)$ assumption in Definition 1.2, we can ensure that each row of the matrix has $k$ elements with a value of 1. This is due to the fact that $\exp(0) = 1$, and $QK^\top$ is a $k$-sparse matrix. Consequently, each row of $B_1$ contains $k$ zero elements. Therefore, $B_1$ is $k$-row sparse.

**Proof of Part 2.** This result can be derived directly from Lemma 5.2, since the output of the classical algorithm is exactly the same as the quantum algorithm.

**Proof of Part 3.** It directly follows from Lemma 7.3. $\square$

## C    OMITTED PROOFS IN SECTION 4

In this section, we provide proofs for the results in Section 4.

*Proof of Lemma 4.3.* The error analysis based on matrices $A$ and $B$ is proved as follows.

**Proof of Part 1.**    It follows that

$$
\begin{aligned}
|A_{i,j} - B_{i,j}| &= |A_{i,j} - 1| \\
&\leq |\exp(-\eta) - 1| \\
&\leq 2\eta
\end{aligned}
\tag{1}
$$

where the first step follows from Definition 4.1, the second step is due to Condition 1 in the statement and the third step is based on simple algebra.

**Proof of Part 2.** For $j \in S_i$, we have

$$A_{i,j} = B_{i,j}. \tag{2}$$

It simply follows that

$$|A_{i,j} - B_{i,j}| = 0.$$

**Proof of Part 3.** For $i \in [n]$, we have

$$
\begin{aligned}
|(A\mathbf{1}_n)_i - (B\mathbf{1}_n)_i| &\leq \sum_{j=1}^{n} |A_{i,j} - B_{i,j}| \\
&\leq \sum_{j=1}^{n} 2\eta \\
&\leq 2n\eta
\end{aligned}
\tag{3}
$$

where the first step follows from triangle inequality, the second step is based on Eq.(1) and Eq.(2) and the third step follows from simple algebra.

**Proof of Part 4.** For $i \in [n]$, we have

$$
\begin{aligned}
(A\mathbf{1}_n)_i &= \sum_{j \in S_i} A_{i,j} + \sum_{j \notin S_i} A_{i,j} \\
&\geq \sum_{j \in S_i} A_{i,j} \\
&\geq \sum_{j \in S_i} \exp(\tau) \\
&\geq k \cdot \exp(\tau) \\
&\geq 2n
\end{aligned}
\tag{4}
$$

where the first step is based on simple algebra, the second step follows from simple algebra, the third step is from Definition 4.1, the forth step is based on the satisfied number, and the last step is from **Condition 1** in the statement.

**Proof of Part 5.** For $i \in [n]$, we have

$$|(A\mathbf{1}_n)_i - (B\mathbf{1}_n)_i| \leq 2n\eta \leq \eta \cdot |(A\mathbf{1}_n)_i|$$

where the first step follows from Eq.(3) and the second step is from Eq.(4). $\qquad\square$

*Proof of Lemma 4.4.* We proof each part below.

**Proof of Part 1.** We have

$$
\begin{aligned}
|D(A)_{i,i} - D(B)_{i,i}| &\leq |(A\mathbf{1}_n)_i - (B\mathbf{1}_n)_i| \\
&\leq \eta \cdot |(A\mathbf{1}_n)_i| \\
&\leq \eta \cdot |D(A)|_{i,i}
\end{aligned}
$$

where the first step is from Definition 1.1, the second step is based on **Part 5** of Lemma 4.3 and the third step is based on Definition 1.1.

**Proof of Part 2.** The error analysis for this part follows a similar approach as in **Part 1**. Due to its similarity, we will omit the details here.

**Proof of Part 3.**   Based on **Part 1.** of Lemma 4.3, for $j \notin S_i$, we have $|A_{i,j} - B_{i,j}| \le 2\eta$. Based on **Part 2.** of Lemma 4.3, for $j \in S_i$, $|A_{i,j} - B_{i,j}| = 0$. Furthermore, taking into account the aforementioned findings, we can conclude that $|A_{i,j} - B_{i,j}| \le 2\eta$. $\qquad\square$

*Proof of Lemma 4.5.*  We first decompose the difference into

$$\|D(A)^{-1}A - D(B)^{-1}B\|_\infty$$
$$\le \|D(A)^{-1}A - D(B)^{-1}A\|_\infty + \|D(B)^{-1}A - D(B)^{-1}B\|_\infty$$
$$= F_1 + F_2. \tag{5}$$

Now, we will provide the upper bounds for $F_1$ and $F_2$, respectively. We have

$$F_1 = \|D(A)^{-1}A - D(B)^{-1}A\|_\infty$$
$$= \max_{i \in [n], j \in [n]} \{|(D(A)^{-1}A - D(B)^{-1}A)_{i,j}|\}$$
$$= \max_{i \in [n], j \in [n]} \{|A_{i,j}(D(A)_{i,i}^{-1} - D(B)_{i,i}^{-1})|\}$$
$$\le \max_{i \in [n], j \in [n]} \{|A_{i,j}| \cdot |\frac{D(A)_{i,i} - D(B)_{i,i}}{D(A)_{i,i}D(B)_{i,i}}|\}$$
$$\le \max_{i \in [n], j \in [n]} \{|\frac{\eta D(B)_{i,i}}{D(A)_{i,i}D(B)_{i,i}}| \cdot |A_{i,j}|\}$$
$$= \eta \cdot \max_{i \in [n], j \in [n]} \{|D(A)_{i,i}^{-1}| \cdot |A_{i,j}|\}$$
$$\le \eta \tag{6}$$

where the first step follows from the definition of $F_1$, the second step is also based on the definition of infinity norm, the third step is due to simple algebra, the fourth step comes from triangle inequality, the fifth step is because of **Part 1.** of Theorem 4.4, the sixth step is due to simple algebra and the last step is due to Definition 1.1.

We have

$$F_2 = \|D(B)^{-1}A - D(B)^{-1}B\|_\infty$$
$$= \max_{i \in [n], j \in [n]} \{|(D(B)^{-1}A - D(B)^{-1}B)_{i,j}|\}$$
$$= \max_{i \in [n], j \in [n]} \{|D(B)_{i,i}^{-1}| \cdot |A - B|_{i,j}\}$$
$$\le 2\eta \cdot \max_{i \in [n], j \in [n]} \{|D(B)_{i,i}^{-1}|\}$$
$$\le 2\eta \tag{7}$$

where the first step comes from the definition, the second step is because of the definition of infinity form, the third step follows from simple algebra, and the last step is from **Part 4** of Lemma 4.3.

By combining the aforementioned findings and conclusions, we can establish the following result.

$$\|D(A)^{-1}A - D(B)^{-1}B\|_\infty$$
$$= F_1 + F_2$$
$$= O(\eta)$$

where the first step follows from Eq. (5) and the second step follows from Eq. (6) and Eq. (7). $\qquad\square$

## D   OMITTED PROOFS IN SECTION 5

*Proof of Lemma 5.1.*  **Proof of Part 1.** For $i \in [n]$, we will focus on a vector $(QK^\top)_{i,*}$.

Given that $j \in [d]$, we define $u(j)$ such that

- $u(j) = 1$ if $(QK^\top)_{i,j} > \tau$

- $u(j) = 0$ else.

According to Part 2 of Theorem 3.1, we can use quantum algorithms to efficiently locate all elements $j \in [n]$ for which $u(j) = 1$.

Give that we compute $u(j)$ in $O(d)$, the time complexity of the quantum algorithm is $\widetilde{O}(\sqrt{nk}d)$.

**Proof of Part 2.**

Based on the proof above, we will output a sparse vector $B_{i,*}$ here. The time complexity of compute the sparse vector can be divided into two parts. One is to find the satisfied element in $\widetilde{O}(\sqrt{nk}d)$, which has been proven above.

Given that $k$ represents the upper bound on the number of satisfied elements, the matrix computation specifically targets those satisfied elements, resulting in a time complexity of $O(kd)$.

The proof is now complete. $\qquad\square$

*Proof of Lemma 5.2.* We have
$$\|D(A)^{-1}AV - D(B)^{-1}BV\|_\infty$$
$$\leq \|D(A)^{-1}BV - D(B)^{-1}BV\|_\infty$$
$$+ \|D(A)^{-1}BV - D(A)^{-1}AV\|_\infty.$$
For each $(i,j) \in [n] \times [d]$, Based on Lemma 4.5, we have
$$|(D(A)^{-1}BV - D(B)^{-1}BV)_{i,j}|$$
$$= |\sum_{l=1}^{n}(D(B)_{i,i}^{-1} - D(A)_{i,i}^{-1}) \cdot B_{i,l} \cdot V_{l,j}|$$
$$\leq \sum_{l=1}^{n}|(D(B)_{i,i}^{-1} - D(A)_{i,i}^{-1}) \cdot B_{i,l}| \cdot \|V\|_\infty$$
$$\leq \sum_{l=1}^{n}|\frac{D(B)_{i,i} - D(A)_{i,i}}{D(B)_{i,i}D(A)_{i,i}}B_{i,l}| \cdot \|V\|_\infty$$
$$\leq \eta \cdot \sum_{l=1}^{n}|D(B)_i^{-1}B_{i,l}| \cdot \|V\|_\infty$$
$$= \eta \cdot |\sum_{l=1}^{n}D(B)_i^{-1}B_{i,l}| \cdot \|V\|_\infty$$
$$= \eta \cdot \|V\|_\infty$$
$$= O(\eta^2) \tag{8}$$
where the first step follows from simple algebra, the second step is based on the definition of infinity norm, the third step is because of simple algebra, the forth step is from Lemma 4.4, the fifth step is because of Definition 1.1, the sixth step is based on Definition 1.1, and the last step is because of $\|V\|_\infty \leq \eta$.

For each $(i,j) \in [n] \times [d]$, we have
$$|(D(A)^{-1}BV - D(A)^{-1}AV)_{i,j}| = |\sum_{l=1}^{n}(D(A)_{i,i}^{-1}(B_{i,l} - A_{i,l}) \cdot V_{l,j}|$$
$$\leq \sum_{l=1}^{n}|(D(A)_{i,i}^{-1}| \cdot |(B_{i,l} - A_{i,l})| \cdot \|V\|_\infty$$
$$= O(\eta^2) \tag{9}$$
where the first step is based on simple algebra, the second step is because of triangle inequality, and the last step is based on $\|V\|_\infty$.

The proof is enhanced by combining Eq. (8) and Eq. (9). $\qquad\square$

# E    OMITTED PROOFS IN SECTION 6

*proof of Theorem 6.1.* The matrix $B$ serves as an approximation matrix for the output of Algorithm 1. The inference process, which relies on the sparsity matrix, is outlined in Algorithm 2.

To analyze the time complexity of the inference stage, we can break it down into two main parts:

**Construction of matrix** $B$: As shown in Lemma 5.1, the time complexity for constructing matrix $B$ is approximately $\widetilde{O}(n\sqrt{nk}d)$.

**Inference with sparsity matrix**: Given the sparsity matrix, as per Lemma 6.2, the time complexity for the inference step is $O(nkd)$. Hence, the overall time complexity for this quantum algorithm in the inference stage is $\widetilde{O}(n^{1.5}k^{0.5}d + nkd)$. With this, we conclude our time complexity analysis. Additionally, based on Lemma 5.2, we can have the following inequality:

$$\|D(A)^{-1}AV - D(B)^{-1}BV\|_\infty \leq O(\eta^2).$$

The theorem is then proved.    □

*Proof of Lemma 6.2.* The computation can be divided into two parts

- Part 1. $D(B)^{-1}MV$.

- Part 2. $D(B)^{-1}(B - M)V$.

**Time Complexity of Part 1.**    It will take $O(nd)$ to compute $MV$. And then, the time complexity of the following step $D(B)^{-1}MV$ is $O(nd)$. The time complexity of the first step is $O(nd)$.

**Time Complexity of Part 2.**    We define

$$C := (B - M).$$

According to statement, for each $i \in [n]$, we have

$$\{j \in [n] \mid C_{i,j} \neq 0\}| \leq k.$$

It will take $O(nkd)$ to compute $\underbrace{CV}_{n \times d}$.

And $D(B)^{-1}CV$ will take $O(nd)$. The second part will need $O(nkd)$.

By combining the conclusions above, the time complexity is $O(nkd + nd) = O(nkd)$.    □

# F    OMITTED PROOFS IN SECTION 7

*Proof of Lemma 7.3.* At the beginning of the algorithm, we will first initialize $\mathcal{M}$ using $\mathcal{M}.\textsc{Init}(B, n, d)$, which has a time complexity of $O(n \log n)$.

**Proof of Part 1.**    And then, we will query $K_i$, using $\mathcal{M}.\textsc{Query}(K_{i,*}, \tau)$ for each $i \in [n]$, which has a time complexity of $\widetilde{O}(n^{2-1/\lfloor d/2 \rfloor} + nk)$ in total.

**Proof of Part 2.**    It will take $\widetilde{O}(n \log n + nk + n^{2-1/\lfloor d/2 \rfloor}) = \widetilde{O}(nk + n^{2-1/\lfloor d/2 \rfloor})$ time to identify the satisfied elements as mentioned above. The matrix is specifically designed to target these satisfied elements, and the remaining steps can be done in a time complexity of $O(nkd)$. The proof is now complete.    □

# G    OMITTED PROOFS IN SECTION 8

*Proof of Lemma 8.2.* We prove a hardness result for an easier task: deciding whether there is at least *one* entry in $QK^\top$ greater than $\tau$.

Suppose there exists a classical algorithm that solves this problem in $n^{2-\epsilon}$-time. Let $A, B \subset \mathbb{R}^d$ be an instance of Max-IP. We construct matrices $Q$ and $K$ using vectors from $A$ and $B$, respectively. Then, we do a binary search for $\mathsf{Max\text{-}IP}(A, B)$. For each candidate value $\tau$, we run the classical algorithm to decide whether there exists an entry with a value greater than $\tau$. Note that the binary search takes $O(\log n)$ rounds. Hence, Max-IP can be solved in $n^{2-\epsilon} \cdot \log n < n^{2-o(1)}$ time, which contradicts the lower bound in Theorem 8.2.

Therefore, no classical algorithm can find all the large entries in $QK^\top$ in $n^{2-\Omega(1)}$-time. $\qquad\square$

## LLM USAGE DISCLOSURE

LLMs were used only to polish language, such as grammar and wording. These models did not contribute to idea creation or writing, and the authors take full responsibility for this paper's content.

