# OpenReview forum: "Quantum Attention: Fast Algorithms for Scalable Computation"
_ICLR.cc/2026/Conference — ICLR 2026 Conference Withdrawn Submission_

### Official Review · Reviewer_kvaf · 2025-10-24

**Soundness:** 1
**Presentation:** 2
**Contribution:** 1
**Rating:** 2
**Confidence:** 4

**Summary:**

The paper proposes a quantum attention algorithm designed to accelerate the computation of the transformer attention matrix. The authors use Grover’s search to identify large entries in the matrix $ QK^\top $ under an assumed $(\tau, k)$-sparsity condition. This substitution is claimed to reduce the classical $O(n^2 d) $time complexity to $ \tilde{O}(n^{1.5}\sqrt{k} d) $. The paper also describes a decomposition of the approximated attention matrix into a rank-1 component plus a sparse component and presents a classical variant using half-space reporting data structures.

**Strengths:**

1. The paper addresses quantum acceleration of large-scale ML models, which could be potentially interesting in principle.
2. The mathematical statements are presented formally with consistent notation.
3. The authors demonstrate awareness of both quantum-algorithm and attention-mechanism literature.

**Weaknesses:**

1. The entire contribution reduces to "use Grover’s search to find large entries of $ QK^\top $", which is essentially a graduate (or advanced undergraduate) homework level exercise. This could be an exceptional undergraduate thesis topic, but I don't think it suits the criteria of ICLR.
2. The paper requires QRAM and sparsity conditions that are not satisfied in practice, unless the authors can demonstrate otherwise.
3. THe paper lacks experiments, simulations, or resource estimates.
4. The paper cannot be connected to practical ML research, and has no realistic quantum implementations.
5. The paper ignores extensive literature on efficient classical attention mechanisms already achieving sub-quadratic scaling.

Overall, I think the authors could consider the paper for quantum machine learning conferences or journals instead of ICLR.

**Questions:**

1. Can the authors provide empirical evidence that real transformer attention matrices exhibit $(\tau, k)$-sparsity?
2. How is the quantum oracle for $ QK^\top $ constructed? How can it be implemented on current quantum computing hardware? and what is the total cost of state preparation and data loading?
3. How does the end-to-end runtime (including QRAM overhead) compare with classical approximate nearest-neighbor or low-rank kernel methods?
4. Would the asymptotic advantage remain if realistic error correction and qubit counts are considered?

**Details Of Ethics Concerns:**

No ethics concerns.

---

> ### Author Response · Authors · 2025-12-01
>
> Thank you for your thoughtful feedback. Your comments are very helpful and much appreciated. We will address these in the next version.

---

### Official Review · Reviewer_71Tk · 2025-10-29

**Soundness:** 2
**Presentation:** 3
**Contribution:** 2
**Rating:** 2
**Confidence:** 4

**Summary:**

The authors propose using Grover search to find elements in the QK^T matrix that exceed a given threshold. Assuming the matrix is sparse, this achieves a quadratic speedup for evaluating the attention mechanism.

**Strengths:**

1. Based on Grover search. Theoretically grounded.
2. The paper is generally well-written.

**Weaknesses:**

1. This is purely theoretical work. The assumptions, such as QRAM, oracle implementation, and qubit count requirements, are either impractical in the near term or not discussed.

**Questions:**

1. How many qubits are required in the proposed method?
2. How can the oracle in line 088 be implemented as a quantum circuit?
3. How can the information of QK^T even be encoded such that the oracle can evaluate the logic? It is likely the complexity is hiding in the construction of this oracle [1].

[1] Phys. Rev. X 14, 041029

---

> ### Author Response · Authors · 2025-12-01
>
> Thank you for your thoughtful feedback. Your comments are very helpful and much appreciated. We will address these in the next version.

---

### Official Review · Reviewer_VYkC · 2025-10-31

**Soundness:** 2
**Presentation:** 3
**Contribution:** 1
**Rating:** 2
**Confidence:** 4

**Summary:**

This paper proposes a quantum algorithm to accelerate the computation of the attention mechanism in Large Language Models (LLMs). The standard attention mechanism has a quadratic complexity $O(n^2 d)$ with respect to the input context length $n$, which is a key scaling challenge.The authors' core idea is to assume that the attention matrix $A = \exp(QK^\top)$ is approximately sparse3. They formalize this with the "($\tau, k$)-good matrix" assumption, which states that for any row $i$, the matrix $QK^\top$ has at most $k$ entries greater than or equal to a threshold $\tau$4.Leveraging this assumption, the paper proposes using Grover's Search algorithm to identify the indices of these $k$ large entries for each of the $n$ rows. The authors claim this search can be performed in $\tilde{O}(\sqrt{nk}d)$ time per row, assuming $O(d)$ oracle cost7. This leads to a total time complexity of $\tilde{O}(n^{1.5}k^{0.5}d + nkd)$ to construct and compute an approximated sparse attention matrix. The authors claim this achieves a polynomial speed-up over the $O(n^2 d)$ classical baseline9.

**Strengths:**

- The paper identifies an important and challenging problem: the quadratic scaling of the attention mechanism in LLMs.

**Weaknesses:**

- The entire quantum speedup hinges on the claim that the Grover oracle $\mathcal{O}_i$ costs $O(d)$-time per query. This $O(d)$ cost is explicitly stated to be "assuming the data are stored in QRAM". QRAM (Quantum Random Access Memory) is a highly speculative, non-trivial, and currently non-existent hardware component. Basing the central claim of a paper on the existence of such a device without any further analysis of this dependency is a severe soundness issue. The paper fails to discuss the implications if QRAM is not available, which would likely make the oracle cost $O(nd)$ or worse, completely negating any quantum speedup.

- The algorithm's utility is entirely dependent on the "($\tau, k$)-good matrix" assumption15. The paper provides no empirical validation for this assumption. It is highly probable that for many attention heads and layers, the attention matrix is not sparse, or that $k$ scales with $n$ (e.g., $k=O(n)$). In the latter case, the quantum algorithm's runtime becomes $\tilde{O}(n^{1.5}\sqrt{n}d) = \tilde{O}(n^2 d)$, which offers no speedup over the classical baseline. A theoretical paper making such a strong assumption must provide evidence of its validity in the target domain (LLMs) or, at minimum, analyze the (poor) performance when the assumption is violated.

-  The $(\tau, k)$-good matrix assumption is foundational to the entire paper, yet it is presented without any empirical validation from actual LLM attention maps. If $k$ is not small, the algorithm provides no speedup.

**Questions:**

1. A complexity analysis for your quantum algorithm without the QRAM assumption. What would be the cost of the Grover oracle $\mathcal{O}_i$ on a more realistic quantum hardware model, and how does this affect the final runtime?

2. Empirical validation for the $(\tau, k)$-good matrix assumption using real-world LLMs (e.g., LLaMa, GPT-2)? Please show how $k$ scales as $n$ (context length) increases across different layers and heads. What is the performance of your algorithm if $k=O(n)$?

3. Please clarify the critical inconsistency between Algorithm 1 (which produces a sparse matrix with 0s) and Definition 4.2 (which defines an approximated matrix with 1s). Which matrix $B$ is actually being used and analyzed? How does this choice affect the proof of Theorem 1.3, specifically the "sparse-plus-rank-one" claim 64and the error analysis in Lemma 4.5?

4. Given that numerous classical approximate attention mechanisms achieve $O(n)$ or $O(n \log n)$ complexity (e.g., Performers, Linformer), why should the research community be interested in a $\tilde{O}(n^{1.5}\sqrt{k}d)$ quantum algorithm that is (a) asymptotically slower and (b) requires speculative, non-existent hardware like QRAM?

---

> ### Author Response · Authors · 2025-12-01
>
> Thank you for your thoughtful feedback. Your comments are very helpful and much appreciated. We will address these in the next version.

---

### Official Review · Reviewer_WfyJ · 2025-11-01

**Soundness:** 3
**Presentation:** 1
**Contribution:** 1
**Rating:** 2
**Confidence:** 4

**Summary:**

This paper proposes using Grover's algorithm to speed up approximate attention computation by assuming that attention matrices are approximately k-sparse. They provide error analysis and complexity analysis to back up their arguments.

**Strengths:**

* I believe the results are correct.
* The paper provides an interesting potential application of Grover's algorithm to machine learning.
* To the best of my knowledge, the idea to apply Grover's algorithm to attention simulation, particularly with the additional assumption of sparse attention matrices, is original.
* The paper may be significant in its potential to inspire new quantum algorithms for machine learning. Additionally, once quantum computers become more practical, this paper could, in principle, provide real-world speedups.
* The overarching idea is clear.

**Weaknesses:**

* The practicality of the proposed method is unclear. In particular, it will require years for quantum computers to be able to run the proposed algorithm at the desired scale, and since the gain is only by a factor of sqrt(nk), classical computers will still likely be faster for a long time.
* Along the lines of the first weakness, the paper could be improved by providing a clearer picture of the intended use of their algorithm. I.e., do the authors envision it as a purely theoretical result or a practical one? If it is a practical result, it would be useful to understand more about the target application. Is the idea to run an entire LLM on a quantum computer, or just the attention (the latter would incur substantial communication costs)? Additionally, it would be useful to compute k and tau for real-world attention matrices to provide more concrete analysis of potential speedups and use cases.
* If it is a theoretical result, it would be useful to frame which deviations from real-world attention are allowable. In particular, why are we constrained to classical attention (and not quantum variants like in Shi et al., 2024) but allowed to make sparsity assumptions? This tradeoff could be more carefully motivated.
* The exposition could be clearer. Although the overarching idea is clear, the paper is laid out in a way that requires a bit of jumping around to follow, causing it to be inconvenient to track down details.

**Questions:**

See weaknesses.

---

> ### Author Response · Authors · 2025-12-01
>
> Thank you for your thoughtful feedback. Your comments are very helpful and much appreciated. We will address these in the next version.

---

### Note · Authors · 2025-12-01

**Comment:**

We would like to sincerely thank all the reviewers for providing insightful feedback. After careful consideration, we have decided to withdraw this paper.

**Withdrawal Confirmation:**

I have read and agree with the venue's withdrawal policy on behalf of myself and my co-authors.